# Egg Yolk Fat Deposition Is Regulated by Diacylglycerol and Ceramide Enriched by Adipocytokine Signaling Pathway in Laying Hens

**DOI:** 10.3390/ani13040607

**Published:** 2023-02-09

**Authors:** Qianyun Ji, Penghui Chang, Yuhao Dou, Yutong Zhao, Xingyong Chen

**Affiliations:** 1Collage of Animal Science and Technology, Anhui Agricultural University, No. 130 Changjiang West Road, Hefei 230036, China; 2Anhui Province Key Laboratory of Local Livestock and Poultry Genetic Resource Conservation and Bio-Breeding, Anhui Agricultural University, No. 130 Changjiang West Road, Hefei 230036, China

**Keywords:** indigenous chicken, commercial laying hen, yolk fat content, yolk fatty acids, liver metabolites

## Abstract

**Simple Summary:**

It is well known that the fat in egg yolk is mainly synthesized by the liver, which is an important organ for the synthesis of lipids in poultry, and then is transported through blood circulation. To investigate the effect of liver lipid metabolism on yolk fat deposition, we measured the quality of egg and yolk, yolk fatty acid composition, and liver lipids in two different chicken breeds. The results suggested that the higher fat content of indigenous chickens was regulated by the upregulated adipocytokine signaling pathway in the liver, which causes enrichment of diacylglycerol and ceramide.

**Abstract:**

The mechanism which regulates differential fat deposition in egg yolk from the indigenous breeds and commercial laying hens is still unclear. In this research, Chinese indigenous Huainan Partridge chickens and Nongda III commercial laying hens were used for egg collection and liver sampling. The weight of eggs and yolk were recorded. Yolk fatty acids were determined by gas chromatography-mass spectrometry. Lipid metabolites in the liver were detected by liquid chromatography-mass spectrometry. Yolk weight, yolk ratio and yolk fat ratio exhibited higher in the Huainan Partridge chicken than that of the Nongda III. Compared to the Nongda III, the content of total saturated fatty acid was lower, while the unsaturated fatty acid was higher in the yolk of the Huainan Partridge chicken. Metabolites of phosphatidylinositol and phosphatidylserine from glycerolphospholipids, and metabolites of diacylglycerol from glycerolipids showed higher enrichment in the Huainan Partridge chicken than that of the Nongda III, which promoted the activation of the adipocytokine signaling pathway. However, metabolites of phosphatidic acid, phosphatidylcholine, phosphatidylethanolamine and lysophosphatidylcholine from glycerol phospholipids, and metabolites of triacylglycerol from glycerolipids showed lower enrichment in the Huainan Partridge chicken than that of the Nongda III. The high level of yolk fat deposition in the Huainan Partridge chicken is regulated by the activation of the adipocytokine signaling pathway which can promote the accumulation of diacylglycerol and ceramide in the liver.

## 1. Introduction

Under intensive selection, egg production has reached more than 300 in commercial egg-type hens within 72 weeks, and could reach 500 in commercial flocks with an extended laying cycle lasting to 100 weeks [1]. However, falling egg quality, including lower shell quality and yolk fat of commercial laying hens has caused the consideration of choosing other types of eggs. The acceptability of eggs is mainly affected by the characteristics which show their intrinsic quality including egg weight, yolk weight, protein content, fat content and the yolk fatty acid composition [2].

Egg quality is mainly dependent on the breed of chickens, feed types, and feed nutrient levels [3], of which breed is considered to be the most important factor affecting egg quality, especially the internal quality of egg yolk fatty acid composition and water content in egg white [4,5]. It is stated that the chicken breed could affect egg quality of the chemical composition of protein, fat (polyunsaturated and saturated), sugar, cholesterol, and α-tocopherol content [6,7]. As egg yolks are rich in nutrients, the inner quality of nutritional composition and content in egg yolk have become the most important factors for consumers to choose eggs. 

The liver is the major organ which is responsible for the synthesis of yolk lipids and other yolk precursors of birds. It is stated that nearly 6 g triacylglycerols are transported to the oocyte from the liver of the laying hens [8,9]. The yolk precursors are mainly composed of very low-density lipoprotein (VLDL) and vitellogenin, which are transported via the vascular system to the oocytes, and enzymatically processed into yolk proteins and deposited in yolk platelets for use [10,11]. During egg laying of Muscovy ducks, the glutathione and ascorbic acid abundance were downregulated and choline abundance was upregulated in the liver to meet the needs of the yolk precursors’ synthesis [12]. During egg laying, phospholipids, triglycerides, apolipoprotein B, and apolipoprotein VLDL-II were all abundantly synthesized in the liver [13], which suggested that the differential chemical components and content from egg yolk might be responsible for the different synthetic ability of yolk precursor in the liver. 

Therefore, this study aimed to investigate the compositional difference between egg yolk and liver metabolism from high-yield commercial egg-type hens of the Nongda III and the indigenous chicken breed of the Huainan Partridge chicken. The result of this experiment will provide new insight to explore the regulatory mechanism of yolk precursor formation and help guide the breeding of laying hens for optimal egg production.

## 2. Materials and Methods

### 2.1. Experimental Materials

Nongda III (commercial laying hens produce pink-colored eggs, CC group) and Huainan Partridge chickens (a Chinese indigenous breed, LC group) at their 40 weeks were selected based on their body weight (1550 ± 100 g) and raised under the same environmental conditions. Birds were housed in 33 × 35 cm laying cages individually with ad libitum access to water and feed under 8 D:16 Lx photoperiods. Nutrient levels during the feeding phase were in accordance with NRC standards. After one week of adaptation, eggs in one day from each breed were collected for quality and yolk fatty acid determination. The laying rate of the commercial Nongda III at 40 weeks of age was 90.6% and 61.2% in the Huainan Partridge chicken. Eight chickens were randomly selected from each group for slaughter and liver samples collection.

### 2.2. Determination of Egg Quality

After weighing the egg, the yolk weight was also recorded for the calculation of the yolk ratio. Yolk liquid was transferred into a centrifuge for fatty acid determination. 

### 2.3. Yolk Fatty Acid Determination and Quantification

The composition and content of fatty acids were analyzed according to Walczak et al. [14], with appropriate modifications. About 1.0 g of yolk sac was weighed and placed into a centrifuge tube together with 0.66 mL internal standard of C11:0 (5 mg·L^−1^). The sample was added with 95% ethanol (0.66 mL) and pure water (1.33 mL) and then ground at 60 Hz for 2 min in a grinder (JX-48, Jinxing, Shanghai, China), and then transferred to a flask containing 33 mg of pyrogallol, 33 mg of zeolite, and 3.3 mL of hydrochloric acid (8.3 mol·L^−1^). After incubation at 75 °C for 40 min, the sample was mixed with 3 mL of 95% ethanol, 10 mL of ether and petroleum ether mixture (V:V = 1:1), and transferred to a separating funnel for standing for 5 min. The ether layer extract was collected into a flask and transferred to a rotary evaporator (RE-2002, Exceed, Shanghai, China) for evaporation till constant weight. The flask was added 1.6 mL of 2% NaOH-methanol solution and incubated in a water bath at 80 °C for 3 min, and then added 1.4 mL of 15% boron trifluoride methanol solution and incubated in a water bath at 80 °C for 3 min. The flask was cooled to room temperature and then mixed with 2 mL of n-heptane and 3 mL of saturated aqueous sodium chloride solution. After standing for 5 min, the upper layer of liquid was transferred to another tube and mixed with 0.6 g anhydrous sodium sulfate. Then, the supernatant was filtered through a 0.22 μm membrane and transferred into a GC injection vial for GC-MS analysis. 

The separated fatty acid methyl ester (FAME) was applied to a DEGS capillary column (DB-WAX, 30 m × 0.25 mm × 0.25 mm, Agilent Technologies Inc., Santa Clara, CA, USA) and analyzed with a flame ionization detector by an Agilent 7980 N gas chromatography (Agilent Technologies, Santa Clara, USA). The initial temperature of the column oven was set at 60 °C for 2 min, followed by heating at 15 °C/min to 230 °C and held for 19 min. The injector and flame ionization detector was maintained at 240 °C, and the injection volume was 1 μL at a split ratio of 10:1. The carrier gas used was nitrogen at a flow rate of 0.8 mL/min. 

Identification of FAME was performed by comparing the retention times to the FAME standards (CDAA-252795, Shanghai Anpu Experimental Technology Co., Ltd., Shanghai, China). The quantity of FAME was analyzed by using C11 as the internal standard according to the formula listed in Walczak et al. [14]. 

### 2.4. Identification of Differential Metabolites in the Liver

#### 2.4.1. Liver Sample Preparation

A liver sample of 60 mg was accurately weighed and transferred into a 1.5 mL centrifuge tube containing 300 μL of methanol-water (V:V = 1:1, containing internal standards GCA-C13, CDCA-D4 and CA-D4) and two small pre-cooled steel balls. The mixed sample was ground for 2 min at 60 Hz in an automatic rapid grinding machine (Jingxin Industrial Development Co., Ltd., JXFSTPRP-24/32, Shanghai, China). The ground mixture was further added with 300 μL chloroform and ultrasonicated for 10 min under an ice bath and then stood for 20 min at −20 °C. After centrifugation at 13,000 rpm of 4 °C for 10 min, the lower chloroform layer was removed into a bottle. The upper layer was added with 300 μL of chloroform-ethanol (V:V = 2:1, containing 0.1 mM BHT), ultrasonicated, and centrifuged again to collect the lower layer into the original bottle. This collected solution was dried in a freeze-concentration centrifugal dryer (Huamei instrument, LNG-T98, Jiangsu, China) and then fully dissolved in 300 μL of isopropanol-methanol (V:V = 1:1). A mixed isotope internal standard of 20 μL was added as the internal standard. After being centrifuged at 13,000 rpm of 4 °C for 10 min, 150 μL of the supernatant was filtered through a 0.22 μm syringe and transferred into an injection vial for ultra-performance liquid chromatography-mass spectrometry (UPLC-MS) analysis. 

#### 2.4.2. UPLC-MS Analysis

The sample was eluted by a gradient elution program with mobile phase A of acetonitrile: water = (60:40, V:V), containing 0.1% FA and 10 mM NH4COOH, and mobile phase B of acetonitrile: isopropanol = (10:90, V:V), composed of 0.1% FA and 10 mM NH4COOH. Mobile phase B was 55% in the first 5 min, and 60% within 6–15 min, then increased to 70% for the next 13 min, 90% for 15 min, then 100% for 16 min. The sample was injected into an ACQUITY UPLC HSS T3 column (100 mm × 2.1 mm, 1.8 μm, Waters) and the injection volume was 5 μL with a flow rate of 0.35 mL/min and a column temperature of 55 °C. 

Chromatographic conditions: ACQUITY UPLC HSS T3 column (100 mm × 2.1 mm, 1.8 μm, Waters Corporation, Milford, MA, USA); elution system consists of mobile phase A acetonitrile: water = (60:40, V:V), containing 0.1% FA and 10 mM NH4COOH, mobile phase B acetonitrile: isopropanol = (10:90, V:V), composed of 0.1% FA and 10 mM NH4COOH. The gradient elution program was as follows: 5 min, 55% B; 10 min, 60% B; 13 min, 70% B; 15 min, 90% B; 16 min, 100% B. The flow rate was 0.35 mL/min, the column temperature was 55 °C, and the injection volume was 5 μL. The mass spectrometry system was the Qtrap5500 mass spectrometry detection system (AB Sciex Company, Framingham, MA, USA) equipped with an electrospray (ESI) ion source and Analyst 1.7 workstation. The mass spectrometry analysis conditions were as follows: curtain gas of 40 psi; ion spray voltage of −4500/5500 V; source temperature at 400 °C; atomizing gas of 50 psi; and auxiliary heating gas of 55 psi. QC samples were injected with four samples analyzed during the whole analysis to provide a repeatable dataset.

#### 2.4.3. Data Processing

The raw data from UPLC-MS analysis was processed by MRMPROBS, an open software for metabolomics data analysis released by Hiroshi Tsugawa’s team at the RIKEN Center for Sustainable Resource Science, for peak extraction, alignment, identification, and area calculation. The metabolites were quantitatively analyzed by the following formula: lipid content (ng/g) = A1/A2 × C × V/N, in which A1 is the peak area; A2 is the peak area of internal standard; C is the concentration of the internal standard (ng/mL); V is the constant volume (0.3 mL); and N is the weight of the sample (g).

### 2.5. Statistical Analysis

The differential metabolites were screened from the two groups by a combination of multivariate statistical analysis and univariate statistical analysis using the raw data being processed by peak extraction, calibration, integration, and normalization, and the Nongda III commercial laying hen was set as the control. The sample stability was analyzed by principal component analysis (PCA) using multivariate statistical analysis. Differential metabolites were screened by partial least squares analysis (PLS-DA), and orthogonal partial least squares analysis (OPLS-DA), which was validated by seven rounds of interactions and 200 response rankings test methods to examine the model quality and prevent model overfitting. Differential metabolites were subjected to KEGG pathway enrichment analysis. The egg quality comparison between the two groups was performed by Student’s *T*-test using SAS 9.3 software, and the data were expressed as mean ± standard error. The differential metabolites were analyzed by using GraphPad prism 8.0.

## 3. Results

### 3.1. Comparison of Egg and Yolk Weight between Commercial Laying Hens and Indigenous Chickens

The egg weight showed no significant difference between commercial laying hens and indigenous chicken breeds (Table 1). However, the yolk weight, yolk ratio, and yolk fat ratio of indigenous Huainan Partridge chickens were significantly higher that of the commercial laying hens (*p* < 0.05).

### 3.2. The Yolk Fatty Acid Composition of Commercial Laying Hens and Indigenous Chickens 

These 11 fatty acids had been detected in the yolk of commercial laying hens and indigenous chickens (Table 2). Compared to the commercial laying hen, the contents of saturated fatty acids (SFA), including C8:0, C12:0, and C18:0, were higher, while the total SFA was lower in indigenous Huainan Partridge chickens. The contents of unsaturated fatty acids (UFA), including C18:1n9c, C18:2n6t, C20:4n6c and total UFA, were all higher in indigenous Huainan Partridge chickens as compared with the commercial laying hens.

### 3.3. Liver Differential Metabolite Analysis

#### 3.3.1. Qualitative Analysis of Liver Metabolites

Liver tissue samples were analyzed by the UPLC-MS platform, and the obtained data were qualitatively analyzed by MRMPROBS software. A total of 693 metabolites were identified, of which 343 were in positive ion mode and 350 were in negative ion mode (Figure 1) as compared with the HMDB, lipid maps, and METLIN databases.

#### 3.3.2. Multivariate Analysis of Liver Metabolites

The system stability was analyzed by principal component analysis (PCA) and QC samples. The QC samples were closely clustered in the PCA model score plot, indicating that the instrument detection was stable and repeatable (Figure 2a).

The main parameter for determining the quality of the PCA model was R2X, and the R2X of PCA was 0.644, which was greater than 0.5, indicating the PCA model was reliable. Similarly, the value of R2X, R2Y, and Q2 of PLS-DA was close to 1, indicating the model could explain and predict the differences between the two samples. The value of Q2 of OPLS-DA was negative, suggesting that the model was not overfitting and had good predictability (Figure 2b). As shown in Figure 2c–e, the LC group and the CC group of the PCA, PLS-DA, and OPLS-DA models were completely separated, respectively, indicating there were large differences in liver metabolites between the two groups of samples. In addition, the metabolites were subjected to the VIP value of the OPLS model for the following analysis.

#### 3.3.3. Analysis of Differential Metabolites

A total of 43 differential metabolites were screened by statistical analysis (Figure 3). The 43 differential metabolites had obvious hierarchical clustering between LC and CC groups (Figure 3a) according to the variable importance in the projection (VIP > 1). To obtain a clear overview of the changes in the differential metabolites, the 43 differential metabolites were further divided into 4 categories: glycerophospholipid metabolites, sphingolipid metabolites, fatty acids, and glyceride metabolites. Each category was divided into one or more types, including phosphatidic acid, phosphatidylcholine, phosphatidylethanolamine, phosphatidylinositol, phosphatidylserine, lysophosphatidylcholine, ceramide, dihydroceramide, sphingomyelin, diacylglycerol, triacylglycerol, and free fatty acid, which all exhibited a significant difference between the LC group and the CC group. 

There were six metabolites categorized into glycerol phospholipid, in which phosphatidylinositol and phosphatidylserine were higher in the LC group, while phosphatidic acid, phosphatidylcholine, phosphatidylethanolamine, and lysophosphatidylcholine were lower in the LC group than those in the CC group (Figure 3b).

There were three metabolites: ceramides, dihydroceramide, and sphingomyelin, which belong to sphingolipid and all exhibited higher in the LC group as compared to the CC group (Figure 3c). Fatty acyls also exhibited higher enrichment in the LC group than in the CC group (Figure 3d).

In the glycerolipids, diacylglycerol had higher enrichment and triacylglycerol had lower enrichment in the LC group than in the CC group (Figure 3e).

#### 3.3.4. Pathway Analysis of Differential Metabolites

Pathway enrichment analysis was performed by using the KEGG IDs of differential metabolites to obtain metabolic pathway enrichment (Figure 4). There were 16 KEGG pathways enriched. The main metabolites-enriched pathways were focused on lipid metabolisms, such as the adipokine signaling pathway, necroptosis, sphingolipid metabolism, glycerophospholipid, and linoleic acid metabolism. The second group of metabolites-enriched pathways was involved in signaling regulation pathways such as ErbB, MAPK, VEGF, and GnRH signaling pathways. The third group of metabolites-enriched pathways was related to others such as the apelin signaling pathway, adrenergic signaling, calcium signaling pathway, and c-type lectin receptor signaling pathway. 

Except for phosphatidylcholine that enriched in glycerophospholipid metabolism and phosphatidylcholine that enriched in linoleic acid metabolism were significantly downregulated, other metabolites enriched in each pathway were all upregulated in the LC group (Table 3). Diacylglycerol (DAG) was enriched in 13 metabolic pathways, involving the adipocytokine signaling pathway, ErbB signaling pathway, MAPK signaling pathway, VEGF signaling pathway, etc. Ceramide (CER) was involved in 3 metabolic pathways, including the adipocytokine signaling pathway, necroptosis and sphingolipid metabolism. Sphingomyelin (SM) was involved in necroptosis and sphingolipid metabolism. Phosphatidylserine (PS) and phosphatidylcholine (PC) were involved in glycerophospholipid metabolism (Table 3).

## 4. Discussion

In layer breeding programs, egg production is an important selection indicator which has had noticeable progress in the genetic selection of commercial laying hens in recent decades. Compared with commercial laying hens, indigenous breeds have a relatively lower egg-laying rate [15]. The chicken breeds with lower egg-laying rate always have the higher levels of liver LDL-C and ovarian LDL-C [16], which might cause a larger amount of yolk precursor (mainly vitelline lipid) deposition in the yolk. As a result, indigenous chickens may have a higher yolk ratio and yolk fat content. Franco et al. found that the contents of fat, protein, and ash in eggs from Mos (a Spanish indigenous breed) were higher than that of Isa Brown [17]. Higher contents of fat and protein and lower content of water in egg yolk was also detected in the Chinese Hotan chicken compared with the Rhode Island Red [18]. Lordelo et al. [19] found that indigenous chickens had a higher proportion of egg yolk by comparing the egg quality of three kinds of indigenous breeds and commercial laying hens. In this research, higher yolk ratio and yolk fat of indigenous Huainan Partridge chickens were higher than that of commercial laying hens which performed a high egg-laying rate.

Yolk fatty acid composition could be largely affected by diet. It is demonstrated that egg yolk SFA and monounsaturated fatty acid (MUFA) concentration were determined by the dietary SFA, MUFA, and 18:2n-6 content [20]. While dietary supplementation with marigolds resulted in increased levels of C16:0 and C18:0 and decreased levels of C16:1 and C18:1 in the egg yolk [21]. However, egg quality, including yolk fatty acids, has been greatly changed after intensive selection of egg production and total egg weight. The content of total SFA, MUFA, and polyunsaturated fatty acids in egg yolk from indigenous chicken breeds was usually higher than those of commercial chickens [19]. Therefore, higher saturated fatty acids and lower unsaturated fatty acids were observed in the crossbred Nongda III commercial laying hens in this research.

Yolk lipids, synthesized by the liver, are mainly composed of triacylglycerol, phospholipid, and free cholesterol, in which triacylglycerol (65%) and phospholipid (32%) are the major components [22]. During laying, fat synthesis is particularly active in the liver of hens. Organisms have the highest and most diverse content of glycerophospholipids, and the various glycerophospholipids can transform mutually [23,24]. Among them, phosphatidylcholine (PC) and phosphatidylethanolamine (PE) could be synthesized through the classical CDP-DAG pathway, while PC, PE, and phosphatidylserine (PS) could be transformed into each other, and PC and PE could directly generate phosphatidic acid (PA) [25], which was the substrate for the formation of diglycerides and triglycerides and could contribute to the formation of lipid droplets [26,27]. A lower enrichment of PC and PE, and higher PS and PI in the liver of Chinese indigenous Huainan Partridge chickens suggested that PC and PE were mostly transformed to the PI and PS to meet more physiological and biochemical functions. In addition, the low enrichment of PC and PE in the liver of indigenous Huainan Partridge chickens and the high content of yolk fatty acids further indicated the transformation of glycerophospholipids metabolites in chickens during laying. Diacylglycerols (DAG) were widely involved in the signal transduction and functional regulation in cells and could form inositol phospholipids, which could cause the change of calcium ion concentration in intracellular, activate protein kinase C and modify many proteins and enzymes in the form of phosphorylation. The inositol phospholipids could be converted from phosphatidic acid under the action of phosphatidylinositol enzyme and promote VLDL assembly and transportation [28,29]. In this study, DAG was upregulated and enriched in multiple metabolic pathways in Huainan Partridge chickens suggesting that lipid synthesis is in high activity so as to cause a high yolk fat content. 

It is suggested that indigenous chicken breeds maintain a higher ability of abdominal fat deposition at their late laying stage [30]. The adipocytokine metabolic pathway and the apelin signaling pathway are directly involved in the body’s fatty acid metabolism. Adipocytokines are a class of biologically active substances synthesized and secreted by adipose tissue to regulate body lipid metabolism [31]. Apelin, as an adipokine, was significantly upregulated in hyperinsulinemia and hyperglycemia mouse models [32]. The activated adipocytokine signaling pathway in the Huainan Partridge chicken indicated a stronger abdominal fat deposition and lower egg production performance. Sphingomyelin (SM) is a component of the cell membrane and participates in important life activities such as apoptosis, cell growth, and differentiation, etc. [33]. Under the catalytic reaction of sphingomyelinase, SM is hydrolyzed to ceramides [34], and the accumulated ceramides inhibit the activity of PKB and produce insulin resistance [35], which causes VLDL synthesis and excessive fat accumulation in the liver [36]. The upregulated sphingolipid metabolism would increase the production of ceramides in the Huainan Partridge chicken and might cause high lipid content in egg yolk.

Laying hens with a body weight of about 1.6 kg need 3 g/d fat from the diet, while the yolk fat content is about 6 g/egg [37]. Therefore, hens need to synthesize more fat to meet the requirement of yolk formation. Under the stimulation of estrogen [38], liver glucose is catalyzed to produce palmitate, which in turn produces long-chain fatty acids and unsaturated fatty acids. The fatty acids synthesized in the liver are subsequently combined with glycerol to form triglycerides and transported to each tissue [39]. A higher enrichment of free fatty acid in the Huainan Partridge chicken suggested its stronger lipid synthesis ability and caused higher yolk fat content.

## 5. Conclusions

Indigenous chickens have a high content of yolk fat which is regulated by the upregulated adipocytokine signaling pathway in the liver, which can promote the enrichment of diacylglycerol and ceramide.

## Figures and Tables

**Figure 1 animals-13-00607-f001:**
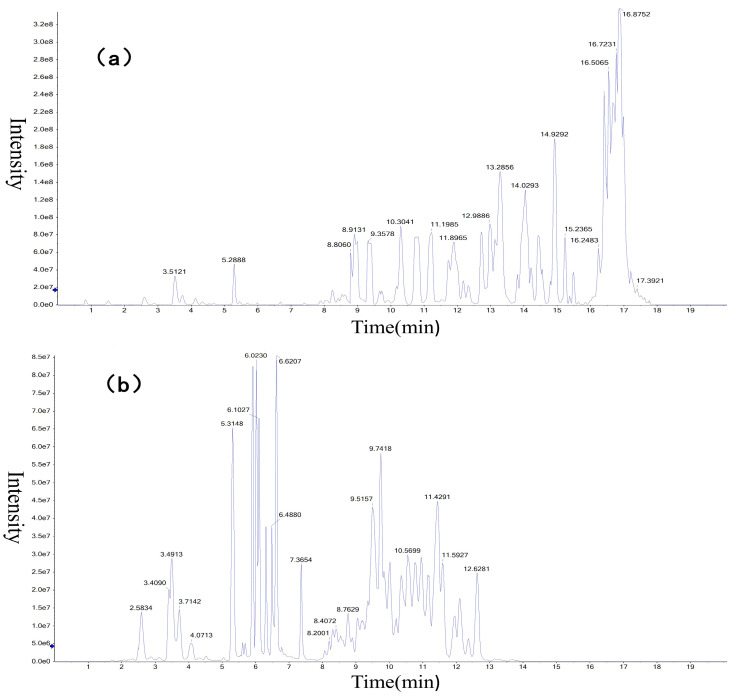
Total ion chromatogram of liver metabolites in (**a**) ES(+) mode and (**b**) ES(−) mode. 8.0 e7 of the ordinate means 8 × 10^7^, the other numbers are the same.

**Figure 2 animals-13-00607-f002:**
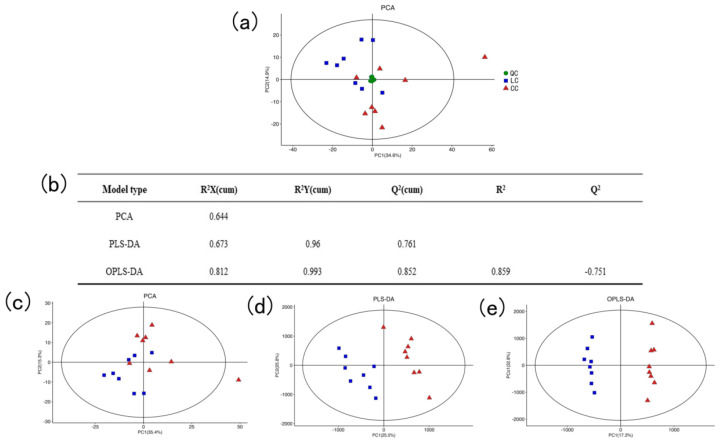
Multivariate statistical analysis. (**a**) PCA score plot for the data from QC samples and all groups. (**b**) The parameter of the statistical model. (**c**) PCA score plot for the data in the LC group and CC group. (**d**) PLS-DA score plot for the data in the LC group and CC group. (**e**) OPLE-DA score plot for the data in the LC group and CC group.

**Figure 3 animals-13-00607-f003:**
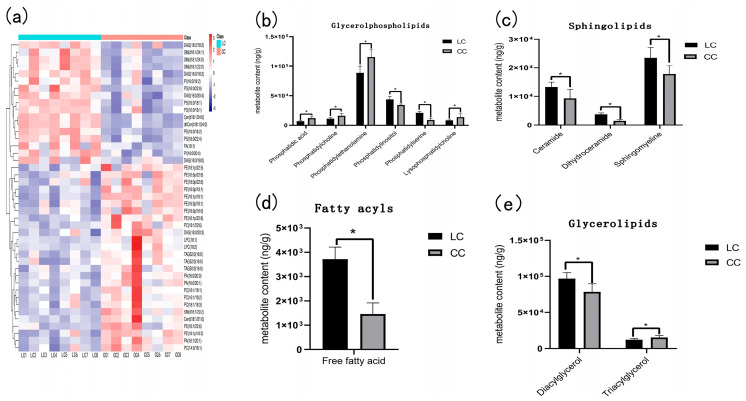
Heatmap and statistical histogram of liver differential metabolites between Huainan Partridge chicken (LC) and Nongda III (CC). * represents the significant difference of *p* < 0.05. (**a**) Heatmap of liver differential metabolites between two groups. (**b**) The content of glycerol phospholipid metabolites in two groups. (**c**) Sphingolipids content of LC and CC groups. (**d**) Fatty acyls content of LC and CC groups. (**e**) The content of glycerolipids metabolites between two groups.

**Figure 4 animals-13-00607-f004:**
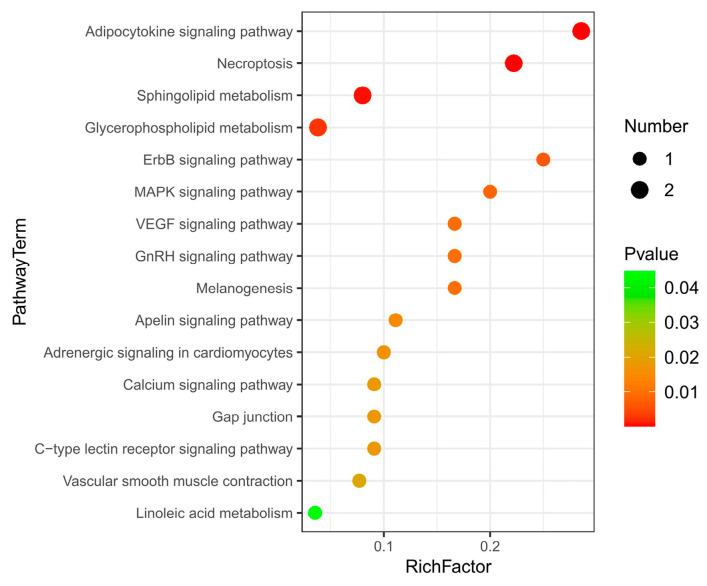
KEGG pathway enrichment of liver differential metabolites between Huainan Partridge chicken (LC) and Nongda III (CC).

**Table 1 animals-13-00607-t001:** Comparison of yolk quality between indigenous chickens and commercial laying hens.

Group	Egg Weight/g	Yolk Weight/g	Yolk Ratio/%	Yolk Fat Ratio/%
LC	56.41 ± 4.92	19.11 ^a^ ± 4.92	33.95 ^a^ ± 2.24	35.37 ^a^ ± 0.71
CC	57.75 ± 5.41	16.74 ^b^ ± 1.98	29.05 ^b^ ± 2.93	27.35 ^b^ ± 0.74

^a, b^ Different letters indicate a significant difference (*p* < 0.05) in a row. CC means commercial laying hens of Nongda III, while LC means indigenous Huainan Partridge chickens. The same as below.

**Table 2 animals-13-00607-t002:** The composition of yolk fatty acid in Huainan Partridge chicken and Nongda III.

Fatty Acid	Retention Time/min	Group		*p* Value
		Group LC	Group CC	
C8:0 (Caprylic acid)	8.012	0.033 ± 0.026	0.003 ± 0.004	0.026
C12:0 (Lauric acid)	11.354	2.230 ± 2.806	0.003 ± 0.002	<0.001
C14:0 (Myristic acid)	12.984	1.848 ± 2.739	2.576 ± 3.148	0.794
C16:0 (Palmitic acid)	15.021	0.106 ± 0.023	0.131 ± 0.031	0.585
C18:0 (Stearic acid)	17.657	0.820 ± 0.944	0.015 ± 0.024	<0.001
ƩSFA	--	0.200 ± 0.089	0.872 ± 1.540	<0.001
C18:1n9t (Elaidic acid)	18.026	0.005 ± 0.003	0.006 ± 0.007	0.183
C18:1n9c (Oleic acid)	18.781	0.026 ± 0.032	0.007 ± 0.009	0.031
C20:1n9c (cis-11-Eicosenoic acid)	22.863	0.508 ± 0.242	0.158 ± 0.084	0.064
C18:2n6t (Linolelaidic acid)	19.305	0.830 ± 1.515	0.250 ± 0.456	0.039
C18:3n6c (γ-Linoleic acid)	20.928	0.031 ± 0.032	0.011 ± 0.018	0.288
C20:4n6c (Arachidonic acid)	27.385	0.066 ± 0.111	0.003 ± 0.003	<0.001
ƩUFA	--	1.283 ± 1.808	0.435 ± 0.519	0.040

**Table 3 animals-13-00607-t003:** Differential metabolites identified in liver from Huainan Partridge chicken and Nongda III.

Annotation ID	Annotation	Metabolites	*p* Value	Up (↑) or Down (↓)
gga04920	Adipocytokine signaling pathway	Diacylglycerol, Ceramide	<0.001	↑
gga04217	Necroptosis	Sphingomyelin, Ceramide	<0.001	↑
gga00600	Sphingolipid metabolism	Sphingomyelin, Ceramide	<0.001	↑
gga00564	Glycerophospholipid metabolism	Phosphatidylserine Phosphatidylcholine	0.003	↑
				↓
gga04012	ErbB signaling pathway	Diacylglycerol	0.006	↑
gga04010	MAPK signaling pathway	Diacylglycerol	0.008	↑
gga04370	VEGF signaling pathway	Diacylglycerol	0.010	↑
gga04912	GnRH signaling pathway	Diacylglycerol	0.010	↑
gga04916	Melanogenesis	Diacylglycerol	0.010	↑
gga04371	Apelin signaling pathway	Diacylglycerol	0.015	↑
gga04261	Adrenergic signaling in cardiomyocytes	Diacylglycerol	0.016	↑
gga04020	Calcium signaling pathway	Diacylglycerol	0.018	↑
gga04540	Gap junction	Diacylglycerol	0.018	↑
gga04625	C-type lectin receptor signaling pathway	Diacylglycerol	0.018	↑
gga04270	Vascular smooth muscle contraction	Diacylglycerol	0.021	↑
gga00591	Linoleic acid metabolism	Phosphatidylcholine	0.045	↓

## Data Availability

The data presented in this study are available on request from the corresponding author.

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
