# Peer review of "Egg Yolk Fat Deposition Is Regulated by Diacylglycerol and Ceramide Enriched by Adipocytokine Signaling Pathway in Laying Hens"

_animals, 2023, doi:10.3390/ani13040607_

Round 1

Reviewer 1 Report

The paper titled Egg yolk fat deposition is regulated by diacylglycerol and ceramide enriched by adipocytokine signaling pathway in laying hens address that high content of yolk fat deposition in Huainan partridge chicken is regulated by the activation of adipocytokine signaling pathway so as to promote the accumulation of diacylglycerol and ceramide in liver. The idea presented herein is currently of interest for indigenous poultry production. However, I think the article needs some improvements to be published.

1. There was no information about the laying rate of two breeds in this article.

2. Line178~282: “Considering chickens at the same age with similar physiological state, the ability of synthesizing yolk precursors could be in similar level. While lower egg laying rate in indigenous breed might cause more yolk precursor deposited in the yolk, which result to a larger yolk and yolk ratio in indigenous chicken breed”. Why not focus on yolk precursors deposition in the ovary?

3. The fat in egg yolk was mainly synthesized by the liver and transported through blood circulation, why not measure the VLDL or vitellogenin in the serum?

4. How many eggs were collected in each breed for quality and yolk fatty acid determination? And how many times the experiment was repeated?

5. Table 2 and Table 3 shared the same title. The title of table 3 was wrong.

6. Line278~282, Line283~285, lack of references.

7. Line75, 85, 88, 117, 118, 135, 157 and 165 should be changed to 2.1, 2.2, 2.3, etc.

8. Too many formatting errors were found in this article. Both  ml and mL were used in Line91~100; Line110, ℃ should be changed to “°C”; Line130~131, 20μl, lack of space between 20 and μl, as well as Line229, 4categories, ect.

9. Revisions of the English written and formatting errors are suggested.

Author Response

Dear reviewer,

     Thank you for your valuable and kind comments. We have reponsed to your comments point-by-point and revised the manuscript according to your comments. Please cheak the attachment for details.

Kind regards,

The author of this manuscript.

Reviewer 2 Report

Overall grammar and language is very poor. Needs to be corrected thoroughly and carefully. Few examples are given below:

Line14-15:  correct the grammar mistake.

Line 16-18: Re-write the sentence with correct grammar.

Line 86: Replace 'after weighing egg weight' with 'after weighing egg'.

Line 101: Replace 80 °C with 80° C. Also check and correct in the whole manuscript accordingly.

Line 99: 0.66 ml internal standard

Line92-94: Correct the sentence and re-write

Line 121: use correct symbol for "comma"

Line 122: Replace 'grounded' with 'ground' Correct in the whole manuscript accordingly.

Line 189: correct the sentence There were 11 fatty acids been detected in yolk from commercial laying"

Line 355: correct the sentence This research was financially by the by the Key Technologies Research and Development Program of Anhui Province

Author Response

(The authors gave the same response as above.)

Reviewer 3 Report

This is a difficult paper to review.  The research appears that it may be strong but I could be wrong.  The writing is, however, very difficult to follow. It appears that the authors are comparing lipid components of eggs in two disparate lines of chickens. The work would be strengthened by inclusion of data from F1 and F2 crosses between the lines.

Tables and figures need to be understandable without reference to the text. Table 2 is titled – “The composition of yolk fatty acid in Huainan partridge chicken and Nongda III”.  Yet it seems to show something else, perhaps, relationships with expression of specific genes.  The figure legend of figure 3 does not fully or even adequately describe the figure.  Parenthetically, I am not clear whether the authors compared the lipid composition of yolk between the two lines in the manner employed with liver lipids (Table 2). 

The authors need to cite where they are getting their information. For instance, in the discussion, they state “the average egg weight about 55 g is preferred for consumers when choose quality eggs.”  Does this reflect a scientific study or the assumptions of the industry?

The manuscript needs to be revised by someone with expertise in both written English and the science before it can be reviewed in detail.

Author Response

(The authors gave the same response as above.)

Round 2

Reviewer 3 Report

The paper is much improved.